# Morphogenesis of liquid crystal topological defects during the nematic-smectic A phase transition

Min-Jun Gim[1], Daniel A. Beller[2] & Dong Ki Yoon[1]

The liquid crystalline phases of matter each possess distinct types of defects that have drawn great interest in areas such as topology, self-assembly and material micropatterning. However, relatively little is known about how defects in one liquid crystalline phase arise from defects or deformations in another phase upon crossing a phase transition. Here, we directly examine defects in the *in situ* thermal phase transition from nematic to smectic A in hybrid-aligned liquid crystal droplets on water substrates, using experimental, theoretical and numerical analyses. The hybrid-aligned nematic droplet spontaneously generates boojum defects. During cooling, toric focal conic domains arise through a sequence of morphological transformations involving nematic stripes and locally aligned focal conic domains. This simple experiment reveals a surprisingly complex pathway by which very different types of defects may be related across the nematic–smectic A phase transition, and presents new possibilities for controlled deformation and patterning of liquid crystals.

[1] Graduate School of Nanoscience and Technology and KINC, KAIST, Daejeon 34141, Republic of Korea. [2] Paulson School of Engineering and Applied Sciences, Harvard University, Cambridge, Massachusetts 02138, USA. Correspondence and requests for materials should be addressed to D.K.Y. (email: nandk@kaist.ac.kr).

Materials composed of anisotropic organic molecules may display a number of mesophases called liquid crystals (LCs), with order and material properties intermediate between fluid and crystalline solids[1]. When the molecular order relaxes after a phase transition or the application of an externally imposed stimulus, there commonly appear defects where the LC order locally breaks down, either as kinetically trapped metastable objects or as components of free energy-minimizing configurations[2]. In most currently used LC-based optoelectronic applications, such as displays and modulators, the presence of defects would inhibit device performance and is avoided[3]. However, once they are well controlled, defects hold the potential to dramatically broaden the applications of LCs, such as in micropatterned surfaces[4,5], microlenses[6,7], vortex beam generators[8,9] and particle manipulation systems[10,11].

Besides these technological applications, LC defects have been a subject of study for over a century as a tool for fundamental scientific studies of topological singularities in physics, ever since the discovery of liquid crystallinity[2]. Topological defects in liquid crystals have been studied in analogy with defects related to symmetry breaking in the early universe (the Kibble–Zurek mechanism)[12] and topological defects in other condensed matter systems, including superconductors[13], superfluids[14] and soft ferromagnets[15]. Liquid crystal defects are often strikingly visible under optical microscopy, facilitating their study in comparison with the smaller or cosmologically larger deformations in the analogous systems mentioned above. Furthermore, LC defects are interesting not just as analogues for other systems but also for a wide variety of unique phenomena arising from their interaction with external fields[16,17] and colloidal particles[18–21].

However, very little is known about the fate of defects during the phase transition between two of the most common LC phases, the nematic (N) and the smectic A (SmA) phases, or how this phase transition determines the final structure of the SmA defect patterns upon cooling. At this phase transition, the broken rotational symmetry of the N phase, characterized by a director field $\mathbf{n}(\mathbf{r})$, is augmented by a further broken translational symmetry along the $\mathbf{n}$ direction. The density becomes modulated, forming equally spaced smectic layers to which $\mathbf{n}(\mathbf{r})$ is normal.

Studying topological defects at phase transitions is especially important in light of the crucial role that such defects play in order–disorder transitions such as the celebrated Kosterlitz–Thouless transition in the two-dimensional XY model[22] and a related theory of melting in two-dimensional solids[23]. The paucity of information about defects at the N–SmA phase transition is partly due to an apparent complete reorganization of the LC order at the transition in some systems, driven by strong anchoring at the boundaries (for example, see Supplementary Fig. 1) in which it is hard to discern a relationship between defects in the two phases. Yet, some recent studies have demonstrated continuous and quantifiable changes in LC defects as the N phase is cooled into the SmA phase; these systems include LC shells[24–28] and samples with micron-scale colloidal inclusions[29,30], as well as thin films with multidirectional rubbing at the substrate[31]. These observations raise important questions about the pathways and history dependence of defect transformations across the N–SmA phase transition, and about the role of these transformations in SmA phase defect pattern formation.

Here, we use films of the LC material 8CB, open to the air above and resting on a water substrate below, to study LC defects during the N–SmA phase transition upon cooling by polarized optical microscopy (POM)[31]. The surface anchoring is strongly homeotropic at the LC/air interface, and strongly degenerate planar at the water interface, creating a hybrid-aligned cell[32]. The choice of water as the lower substrate provides a useful and striking new view on the phase transition, as this system provides a uniform thin LC thickness with no azimuthal anchoring potential and a minimal thermal gradient. Because the surface tension dominates over liquid crystalline anchoring energies, the LC/water interface is not significantly perturbed from flatness by director distortions or defects. We observe a dramatic sequence of morphological changes in the director field geometry during the phase transition that reveals surprising connections between the initial N geometry and the subsequent defect configurations[32] (Figs 1 and 2, Supplementary Movie 1). Boojums at the water surface in the N phase (Fig. 1a) become the organizing centres for stripe undulations in the N director $\mathbf{n}$. These stripes break up into

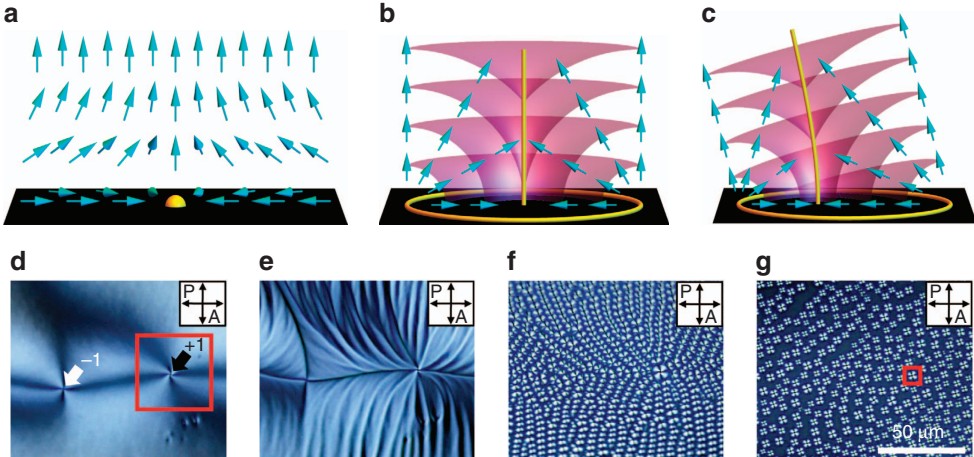

**Figure 1 | Typical defect structures of N and SmA phases. (a)** Schematic diagrams of molecular configuration surrounding boojum at NLC phase in square-outlined area of **d**. The yellow sphere is a surface point defect (boojum). **(b)** Schematic diagram of a TFCD in the SmA phase in square-outlined area of **g**. The yellow lines are the TFCD focal curves, the purple surfaces represent smectic layers and the cyan arrows represent the director. **(c)** Schematic diagram of an elliptic–hyperbolic FCD with nonzero eccentricity e. **(d–g)** Polarized optical microscopy observations of morphological changes of ±1 surface defects during the N to SmA phase transition. **(d)** −1 and +1 defects (white and black arrows) showing four dark brushes spontaneously appear at the NLC phase temperature (33 °C). **(e)** Stripe patterns appear at the vicinity of SmA phase temperature (32.3 °C). **(f)** After the transition to SmA phase, the stripes are divided into small domains (FCDs) having line singularities (32.1 °C). **(g)** FCDs are replaced by scattered TFCDs after further decrease in temperature (31.5 °C). The black area between TFCDs is due to homeotropically aligning LC molecules.

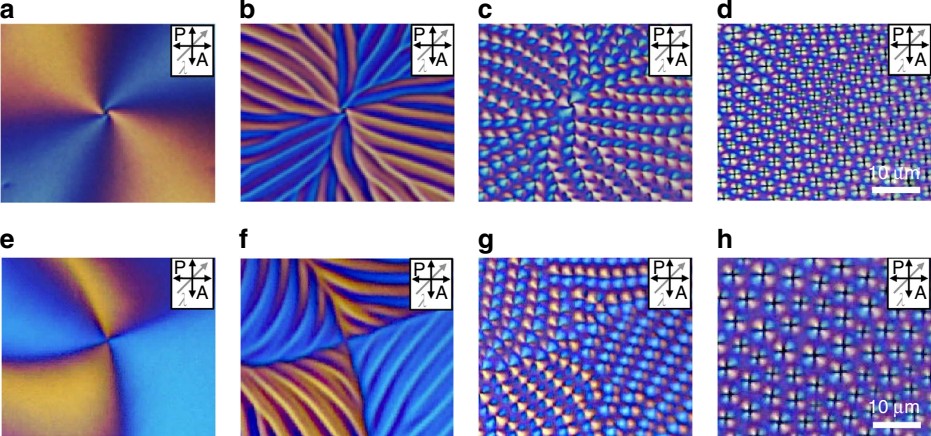

**Figure 2 | POM images of $\pm 1$ boojums under a retardation plate during the phase transition.** (**a,e**) $\pm 1$ boojums at N phase. (**b,f**) Stripes. (**c,g**) FCDs, with one TFCD at the site of the former $+1$ boojum. (**d,h**) TFCDs. Gray arrows in inset of images indicate the direction of slow axis of the retardation plate.

rows of aligned focal conic domains (FCDs), well-known defects of the SmA phase, with nonzero eccentricity and orientations determined by the initial boojum patterning, recalling their N history. Finally, the FCDs relax into toric FCDs (TFCDs) (Fig. 1b)[31,33]. Meanwhile, we observe the direct transformation of certain special N boojums into large TFCDs early in the cooling process. We rationalize our findings through geometrical modelling, along with a theoretical and numerical study of the stripe instability. These results shed significant light on the self-assembly pathway of defects in LC phases, revealing unexpected intermediate states with a strong history dependence, and opening new avenues for potential applications beyond LC-based displays.

## Results

**Morphological changes during the phase transition.** When a drop of 8CB is placed on a water surface with a temperature in the isotropic range of 8CB, the LC drop spreads until achieving a uniform thickness. The drop is then cooled to the N phase in which the molecules are subjected to the antagonistic boundary conditions of hybrid alignment[32]. From top to bottom of the LC film, the director field must rotate by 90° from vertical to horizontal to satisfy these boundary conditions. However, the director's azimuthal angle in the plane of the water substrate is a spontaneously broken continuous symmetry, a unit vector field analogous to the XY model of magnetism and the $c$-director of the smectic C phase. As in those cases, vortices may arise in the projected director field. In the full three-dimensional director field, these vortices are boojums[34,35], point defects at the LC/water interface (Fig. 1a). They arise spontaneously upon cooling from the isotropic phase into the N phase, and are clearly visible as the intersection of four dark and four bright brushes under crossed polarizers (Fig. 1d). In such schlieren textures, the number of dark brushes meeting at a point is four times the absolute value of the winding number of the defect $|s|$[2,32]. The sign of $s$ can be experimentally determined by rotating the crossed polarizers; when the brushes rotate in the same or opposite direction as the polarizers do, the sign is positive or negative, respectively (black or white arrow in Fig. 1d)[36].

Upon cooling sufficiently below the N–SmA transition temperature $T_{NA} = 32.2\,°C$, we observe the TFCD arrays (Fig. 1b,g) that are typical of SmA films under hybrid anchoring[4–6,11,31]. A TFCD consists of a family of smectic layers shaped as nested tori, with a circular defect line at the centre of the tori and a vertical cusp line in the centre (Fig. 1b). The SmA ground state under hybrid anchoring is typically an

array of TFCDs, each bounded by a vertical cylinder as in Fig. 1b, with the interstices between TFCDs filled by planar, horizontal layers. The TFCD is a special, zero-eccentricity case of the FCD, in which the smectic layers take the form of surfaces known as Dupin cyclides, and the defect lines are an ellipse and a hyperbola[33] (Fig. 1c). The FCD is said to have an eccentricity $e$, $0 \le e < 1$, equal to the eccentricity of its elliptical defect, $\sqrt{1 - a^2/b^2}$, where $a$ and $b$ are the lengths of semi-major and semi-minor axes, respectively. FCDs of nonzero eccentricity are often called 'elliptic–hyperbolic FCDs'.

Our study reveals two intricate intermediate states during the cooling process between nematic boojums and smectic TFCDs. First, the nematic director **n** develops stripe undulations, with the stripes running parallel to the initial **n** at the water substrate (Fig. 1e). Then, as the temperature falls below $T_{NA}$, these stripes break up into rows of FCDs (Fig. 1f). Such FCDs have an orientation given by the long axis of their elliptical base, or equivalently by the projection of their hyperbolic cusp defects onto the horizontal plane. We observe that the orientations of the FCDs follow the orientations of the stripes that precede them, which are themselves inherited from the boojum arrangement in the N phase. Only after further cooling do the FCDs relax into TFCDs, with zero eccentricity and azimuthal symmetry (Fig. 1g).

To more precisely determine the director field around the defects during the phase transition, we insert a first-order retardation plate ($\lambda = 530\,nm$) between the sample and the analyser (Fig. 2). The sample then appears magenta where **n** is parallel, perpendicula, or vertical with respect to the polarizers. Cyan-blue and yellow colours appear where **n** is parallel and perpendicular, respectively, to the slow axis of the first-order retardation plate[37]. In the N phase, the region around a boojum consists of two large yellow areas and two large blue areas, all meeting at the boojum site (Fig. 2a,e). In the stripes regime (Fig. 2b,f), the predominant colour in each region is the same as in the boojum regime, but the colour oscillates over the stripe wavelength. This implies large-angle azimuthal undulations of **n** in the direction transverse to the stripes, oscillating about the locally average direction parallel to the stripe direction, inherited from the boojum regime. In the case of Fig. 2b, the stripes are oriented radially outward from the initial boojum. The stripes arise as a result of the diverging ratio of the bend and splay elastic constants, $K_3/K_1$ (refs 38,39), as the N phase is cooled into the SmA phase; we will investigate this phenomenon theoretically and numerically below.

Upon further cooling below $T_{NA}$, FCDs appear throughout the system (Fig. 2c,g). These FCDs at first have a high eccentricity, as

evidenced by the asymmetry of the yellow and cyan-blue lobes of each domain about the point where they meet, in contrast with the symmetric arrangement of the four lobes of the TFCDs (Fig. 2d,h). The lobes also reveal the orientation of each FCD that in the case of Fig. 2c is generally radially inward toward the site of the initial boojum. A closer comparison of Fig. 2b,c reveals that the FCDs are arranged in rows parallel to the domain orientations, and that these rows are the same as the stripes of Fig. 2b. Therefore, the N–SmA phase transition here involves the breaking up of stripes along their lengths into rows of FCDs oriented along the stripe direction, with the minor axis length of the FCDs equal to the stripe wavelength. Even the dislocations of the stripes in Fig. 2b persist as dislocations in the FCD rows of Fig. 2c, creating FCD 'defects of defects'. Finally, further cooling causes a relaxation of the FCD array into an array of TFCDs, with the eccentricity diminishing to zero continuously.

The boojums of the N phase have a winding number of either +1 or −1 in the plane of the substrate, describing the sense of rotation of the director field on a loop around the boojum. The stripes and FCD rows are organized radially around a +1 boojum (Fig. 2a–d). In the case of the −1 boojum, which involves a hyperbolic director field in the N phase (in the plane of the substrate), the stripes and FCDs follow a diamond-shaped pattern around the defect site (Fig. 2e–h)[28]. Because the stripes form parallel to the N phase's horizontal **n** component, the radial or diamond arrangement of stripes inherits and reveals the +1 (radial) or −1 (hyperbolic) windings of the original boojum configuration.

We also observe the direct transformation of a special class of boojums, which we term +1 converging, into TFCDs. In the hybrid-aligned N, the +1 boojums can be divided into converging and diverging types (Fig. 3). While it is well known that the nematic symmetry **n** = −**n** generally forbids such a choice of orientation to be made consistently[3], here the lack of half-integer defects in our system makes it possible[32], and useful, to describe **n** as a unit vector field. In traversing from the air interface to the substrate, the director rotates by 90° to point either inward (converging) or outward (diverging), as shown schematically in Fig. 3b,d–f, where we adopt the convention that the director points up at the air interface. The +1 converging

boojum is frequently referred to in the literature as 'hyperbolic', and the +1 diverging boojum as 'radial'. The converging/diverging distinction has a clear physical manifestation in the SmA phase: the FCDs are oriented towards the boojum site in the converging case, meaning that the hyperbolic focal curve faces the boojum site, with POM showing lobes positioned on the radially outward side (Fig. 3a). In the diverging case, the FCDs are oriented away from the boojum site, with the lobes positioned on the radially inward side (Fig. 3c). The reason the FCD orientations reveal the rotation sense of the N director in the vertical cross-section will be discussed below, in the section 'Geometrical discussion of the stripe–FCD transition'.

It is the +1 converging boojums that transform directly into TFCDs (Figs 2a and 3a) during the breaking up of the stripes into eccentric FCDs. This is because the TFCD, like the +1 converging boojum, has azimuthal symmetry and a converging director field from top to bottom (Fig. 1b). These early-forming TFCDs are larger than the surrounding FCDs, but the size distinction is gradually lost as the FCDs evolve into TFCDs upon further cooling. The +1 diverging boojums, on the other hand, do not evolve directly into TFCDs (Fig. 3c). The −1 boojums also cannot evolve directly into TFCDs because they lack the azimuthal symmetry of TFCDs, as revealed by the diamond-like configurations of the stripes parallel to the original **n** field (Fig. 2f).

**Theory of the nematic stripe instability**. Stripe instabilities in N have a rich and diverse history in the study of LCs, both deep in the N phase and close to the N–SmA transition. In contrast to previous studies[32,38–46], we observe a stripe instability in 8CB on a water substrate upon cooling towards $T_{NA}$ in which the LC thickness is micron-scale and the boundary conditions are those of the hybrid-aligned N (HAN). The planar-anchoring substrate is 'degenerate' planar unlike in ref. 47, while the proximity to $T_{NA}$ makes the bend elastic constant $K_3$ far greater than the splay elastic constant $K_1$. As we will see, stripes arise to decrease the bend distortion of the nematic director field[48].

To develop a theoretical understanding of the stripe instability in our system, we begin with the uniformly distorted HAN

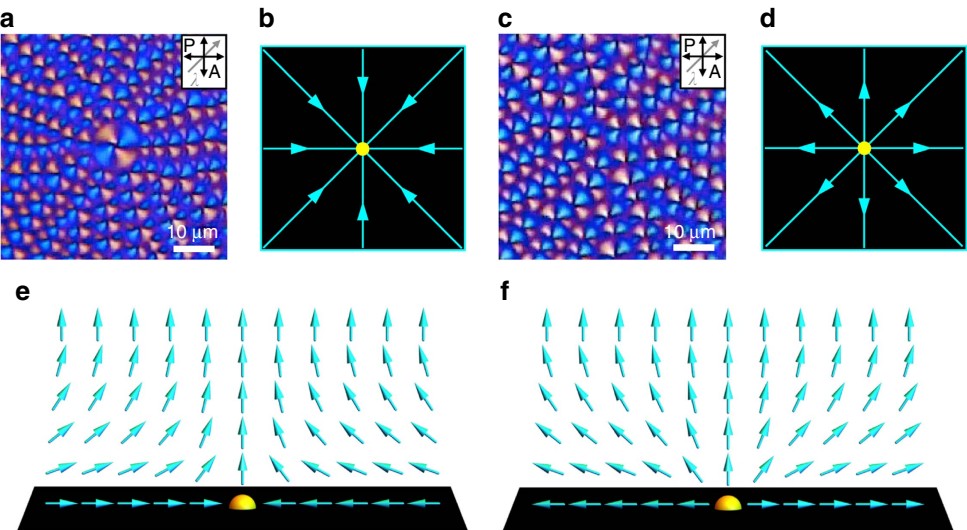

**Figure 3 | Enlarged cores region of two distinguishable +1 defects in the SmA phase.** FCDs orient inward (**a**) and outward (**c**) from different +1 surface defects (POM images using a first-order retardation plate with slow axis along the gray arrow). Schematic diagrams shows convergence (**b**) and divergence (**d**) of direction of FCDs (cyan arrows) in **a**,**b**, respectively. For the +1 boojum defects of the N phase, schematic diagrams of a cross-section of the director field are shown for the converging (**e**) and diverging (**f**) cases. Cyan arrows represent the director with a consistent choice of orientation, and yellow spheres are surface point defects (boojums).

                                                                                                                          

director field before the stripes appear, and we perform a linear stability analysis to predict when stripes become energetically favourable. The details of the calculation can be found in the Methods section. The elastic energy of deformations is calculated using the Frank free energy

$$F_{\text{elastic}} = \frac{1}{2} \int dV \left\{ K_1 (\nabla \cdot \mathbf{n})^2 + K_2 (\mathbf{n} \cdot (\nabla \times \mathbf{n}))^2 + K_3 ((\mathbf{n} \cdot \nabla) \mathbf{n})^2 \right.$$
$$\left. - 2 K_{24} \nabla \cdot [\mathbf{n} (\nabla \cdot \mathbf{n}) - (\mathbf{n} \cdot \nabla) \mathbf{n}] \right\} . \tag{1}$$

The distortion modes in the integrand are respectively the splay, twist, bend and saddle-splay modes. The water substrate contributes a degenerate planar anchoring potential $F_{\text{anchoring}} = \frac{1}{2} W \int dA (\mathbf{n} \cdot \hat{\mathbf{z}})^2$. We assume infinitely strong anchoring at the homeotropic air interface, so that $\mathbf{n}$ is strictly vertical there, which is probably appropriate close to $T_{\text{NA}}$ where homeotropic anchoring strengths increase by orders of magnitude[38]. As the transition temperature $T_{\text{NA}}$ is approached from above, the ratio $K_3/K_1$ of bend to splay elastic constants diverges. In the limit of large $K_3/K_1$, the HAN director field has a polar angle profile $\theta_0(z)$ similar to that shown in Fig. 6a. The director's azimuthal angle $\phi$ is taken to be zero everywhere.

Then, as a function of $K_3/K_1$, we test whether the total energy can be decreased by small sinusoidal undulations of wavelength $\lambda$ in both $\theta$ and $\phi$. We find that energy-decreasing stripes do exist when $K_3$ is greater than a critical value $K_3^{\dagger}$ but less than $HW$ where $H$ is the LC thickness. The stripes must have a wavelength $\lambda$ above a stability boundary $\lambda_*$ as shown in Fig. 4a. The smallest possible wavelength $\lambda_*$, is typically of the same order as the LC thickness. The critical bend elastic constant $K_3^{\dagger}$, above which stripes are expected, depends strongly on both the anchoring strength $W$ and the saddle-splay elastic constant $K_{24}$, as shown in Fig. 4b.

The energetic stability of stripes comes from the bend term in $F_{\text{elastic}}$. This is seen in Fig. 4c, where the stripe wavelength is chosen to be 1.01 times $\lambda_*$. For each of the elastic and anchoring energy terms, Fig. 4c plots its second derivative with respect to stripe amplitude $A$. All terms are positive, indicating an energy cost, except for the bend term that is negative. Note that the reference configuration, the uniform HAN state, contains considerable splay and bend distortions; the stripe undulations form to relieve the increasing cost of bend distortions by shifting elastic energy into all of the other distortion modes. As shown in Fig. 4d, the savings in bend energy is sufficient to give the total energy a negative second derivative with respect to $A$, meaning that some nonzero stripe amplitude will minimize the total energy. Illustrations of such sinusoidal stripes in the HAN director field are shown in Fig. 5a,b,e.

We also perform Landau-de Gennes numerical modelling[49] of the HAN system at high $K_3/K_1$, the results of which validate the above linear stability analysis and yield further insights into the nematic stripe structure. In this approach, the nematic state is modelled as a 3-by-3 Q-tensor defined on a two-dimensional square lattice, from which the director field $\mathbf{n}$ (a unit vector in three dimensions) is extracted as the eigenvector with the greatest eigenvalue. Details of the numerical technique are given in the Methods section.

The numerically calculated HAN director fields show the presence of stripe deformations over an interval of $K_3/K_1$ values similar to that predicted by linear stability theory, as shown in Fig. 4a. Moreover, Fig. 4a shows that the numerically calculated stripe wavelengths $\lambda$ are generally quite close to the theoretically determined stability boundary $\lambda_*$. Thus, our linear stability analysis, while only meant to predict the onset of the stripe instability, turns out to provide a reasonably good prediction for

the energy-minimizing stripe wavelength after the instability. Furthermore, the calculated stripe wavelengths are typically comparable to the film thickness, a result that agrees with the experimental situation and that probably contributes to FCD formation from the stripes, since the semimajor axis length of an FCD cannot exceed the film thickness.

The numerically calculated director field with stripes is illustrated in Fig. 5c,d. The azimuthal and polar angles of $\mathbf{n}$ are plotted in Fig. 5f,g as a function of the coordinate $y$ transverse to the stripes. Comparing Fig. 5e,f, we see that in the midplane $z = H/2$, the numerically calculated $\phi$ and $\theta$ both oscillate sinusoidally, with relative phase and magnitude similar to those expected from the linear stability calculation. (The wavelength is slightly larger in the numerical results, as seen in Fig. 4a.) However, Fig. 5g shows that at the water substrate, the numerically calculated $\phi$ and $\theta$ do not oscillate as simple sine waves; multiple Fourier modes are superposed. In particular, $\phi$ alternates between slow increases and sudden decreases, resulting in the qualitative dissimilarities between Fig. 5b,d. This pattern allows splay distortions to take up more space and confines the costlier bend distortions to smaller regions. The sudden decreases in $\phi$ are the precursors to curvature walls in the smectic phase between FCD rows, while the slow increases accommodate the FCD structure, as discussed below.

**Geometrical discussion of the stripe–FCD transition.** Why do the nematic stripes evolve into elliptic–hyperbolic FCDs at the N–SmA transition? We propose that such FCDs, in particular incomplete FCDs[50,51], provide a field of smectic layer normals in close agreement with the director field of the N stripes, from which the FCDs evolve. Incompleteness of the FCDs refers to the fact that the domains are generally missing a portion of the elliptical focal curve on the side that the hyperbola faces, and a corresponding volume of the FCD is missing as well. This missing volume is occupied by the next FCD in the row. We know the FCDs to be incomplete in part because the bright lobes seen in POM appear only behind the convergence point of the lobes, marking the hyperbola location (Fig. 2c,g), whereas a complete FCD would have small lobes on the forward side as well. In addition, bright-field optical microscopy directly reveals incomplete elliptical defect lines (Supplementary Fig. 2).

A schematic illustration of the system geometry is presented in Fig. 6b, with the incomplete FCDs occupying portions of vertical circular cylinders[52]. The asymptotic direction of the upper portion of the hyperbola is then along the vertical, so that the layers at the air interface are mostly horizontal, smoothly joining onto horizontal layers in the interstices[52,53]. This geometry requires the ellipses to be somewhat tilted out of the plane of the substrate, and a small portion of the FCD under the ellipse is contained within a tilted circular cylinder[53]. Degenerate planar anchoring at the substrate is not strictly satisfied, but this energetic sacrifice is not unusual for hybrid-aligned systems with stronger anchoring at the homeotropic interface[53]. Our optical microscopy measurements show that the hyperbola, viewed from above, appears to be oriented opposite to the FCD orientation, as Fig. 6b would suggest. Incompleteness of an FCD requires curvature walls or many dislocations between neighbouring FCDs, but the energetic cost of such smectic defects may not be prohibitive near $T_{\text{NA}}$ (ref. 51). As the defect cost increases upon cooling deeper into the smectic phase, the FCDs relax into complete (or more nearly complete) TFCDs, with their circular focal curves resting on the substrate. Complete TFCDs join smoothly to one another and to interstitial regions of horizontal layers.

Comparing a side view of the HAN director field profile (Fig. 6a) with the polar angle of the smectic layer normal in the

                                                        

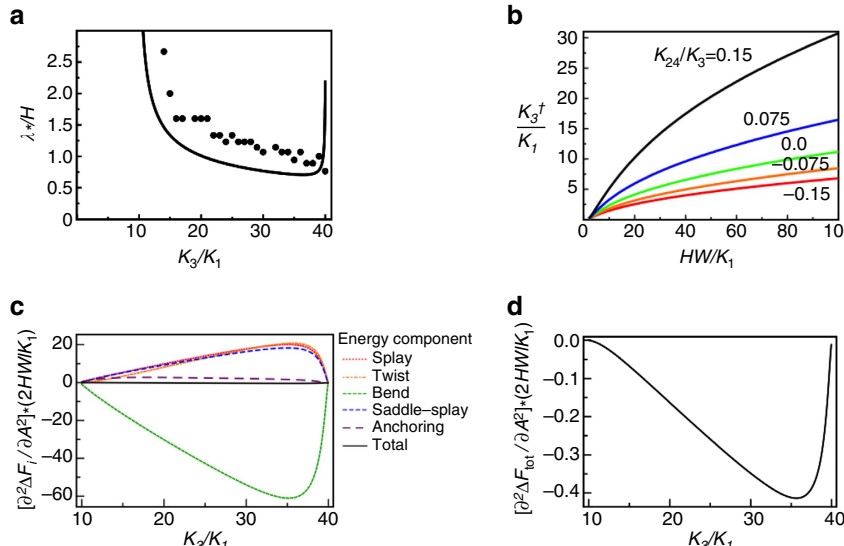

**Figure 4 | Nematic stripe instability in theory and numerics.** (**a**) Stripe wavelength as a function of the ratio of bend and splay elastic constants. Data points are stripe wavelengths obtained using Landau-de Gennes numerical modelling. The curve is the stability boundary $\lambda_*/H$ from linear stability theory. Parameters used are $h = HW/K_1 = 40$, $K_2/K_3 = 0.15$, $K_{24}/K_2 = 0.5$. (**b**) Critical ratio of bend and splay elastic constants at the onset of the stripe linear instability, as a function of $h$. The saddle-splay elastic constant $K_{24}$ may strongly affect the critical ratio; curves from top to bottom are calculated with $K_{24}/K_3 = 0.15, 0.075, 0, -0.075, -0.15$. (**c**) Second derivative of free energy components and total with respect to stripe amplitude $A$, assuming a stripe wavelength of 1.01 times the critical wavelength $\lambda_*$ (just inside the region of stripe instability), using the same parameters as in **a**. The bend term is the only negative term, and drives the instability. (**d**) The total free energy curve from (**c**), with the vertical axis rescaled.

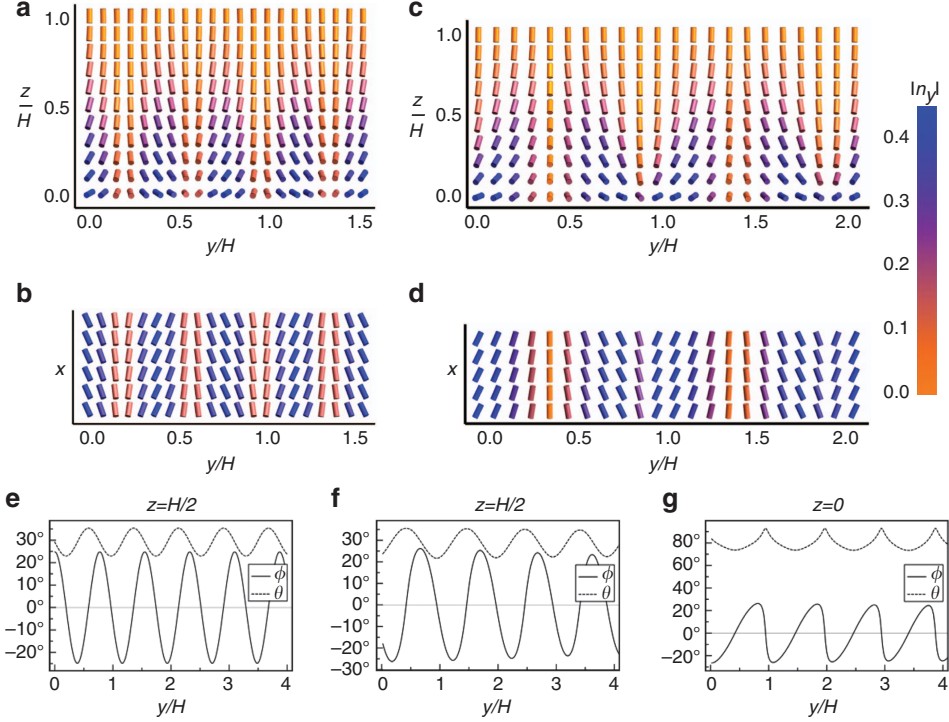

**Figure 5 | Nematic stripe structure in theory and numerics.** (**a,b**) Theoretical nematic stripe director field in (**a**) the $yz$ plane and (**b**) the $xy$ plane at $z = 0$, with a stripe wavelength of 1.01 times the critical wavelength $\lambda_*$ (just inside the region of stripe instability) using the parameters $h = HW/K_1 = 40$, $K_2/K_3 = 0.15$, $K_{24}/K_2 = 0.5$. The arbitrary stripe amplitude is chosen to match the numerical results for illustrative purposes. (**c,d**) Numerical Landau-de Gennes calculation of the director field in (**c**) the $yz$ plane and (**d**) the $xy$ plane at $z = 0$, using the same parameters as in **a,b**. The colour legend at right applies to the $y$-component of the director in **a–d**. (**e–g**) Director field azimuthal angle $\phi$ and polar angle $\theta$ plotted against $y$ for (**e**) the theoretical director field of (**a,b**) at $z = H/2$, (**f**) the numerical director field of (**c,d**) at $z = H/2$, and (**g**) the numerical director field of (**c,d**) at $z = 0$. While (**f**) shows $\theta$ and $\phi$ oscillating sinusoidally with relative amplitudes similar to those shown in (**e,g**) reveals that higher Fourier modes superpose at the substrate: $\phi$ drops suddenly from its maximum to its minimum value, a precursor to curvature walls in the smectic phase.

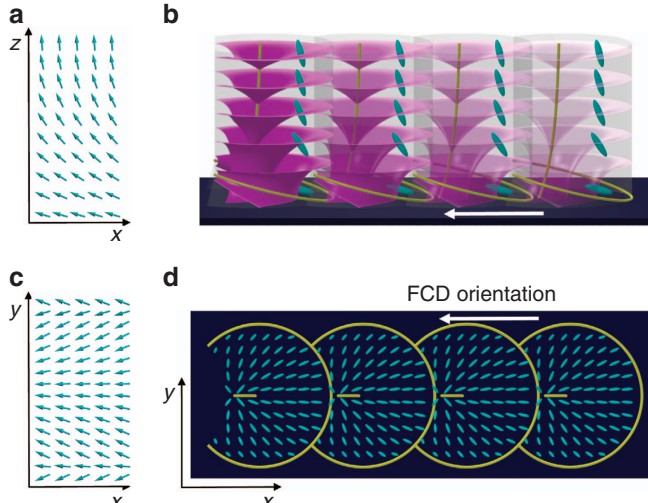

**Figure 6 | Geometry of the stripe–FCD transition. (a)** Side view of numerically calculated hybrid-aligned nematic director field in Fig. 5c,d, with an orientation chosen such that **n** points up at the air interface. **(b)** Illustration of a row of incomplete FCDs with eccentricity $e = 0.3$ (smectic layers in purple, defect lines in yellow) with representative rodlike molecules along the layer normal in cyan. **(c)** Top view of numerically calculated director field, showing precursors to curvature walls at the top and bottom. **(d)** Illustrated top view of the director field in the base of the same row of FCDs as in **b**.

elliptic–hyperbolic FCDs behind the hyperbola (Fig. 6b), we see that the rotation of the director from the homeotropic air interface to the degenerate planar substrate is qualitatively the same in both cases, provided that the FCD orientation agrees with the N in-plane orientation before the stripe onset. On the forward side of the FCDs, which is the side faced by the hyperbola, the director would rotate in the sense opposite to the HAN **n** profile. However, the combination of high eccentricity and incompleteness of the FCDs removes most of this volume.

Meanwhile, in the plane of the substrate, the FCD layer normal direction splays out behind the hyperbola (Fig. 6d) in a way similar to the numerically calculated nematic director field at the substrate (Fig. 6c). This suggests that the structure of the stripes is connected with FCD formation: the monotonic variation of $\phi$ over most of the stripe period (Fig. 5d,g) allows **n** to evolve easily into the rear portions of the FCDs (hence incomplete), while the quick jump in $\phi$ in the small remainder of the stripe period seeds a curvature wall between adjacent FCD rows. Again, the forward side of the FCDs would have an in-plane director field opposite to the N stripe undulation, but high eccentricity and incompleteness of the FCDs remove most of this problematic portion. Thus, rows of incomplete elliptic–hyperbolic FCDs, with the FCDs oriented along the row direction, provide a natural smectic successor structure to the N stripe state, with a very similar director field. This allows the N stripe pattern, organized around the original boojums, to persist in the complex arrangement of the FCDs.

The exact structure of the curvature walls, between FCDs within a row and between adjacent FCD rows, is likely to involve many dislocations and is beyond the scope of this work both experimentally and theoretically. However, it is reasonable to assume that the curvature walls become increasingly energetically costly as the temperature is lowered deeper into the SmA phase and the smectic layer compression modulus increases[1,33,52]. This energetic cost, and the resulting tendency to anneal away the curvature walls as the sample is cooled, drives the continuous evolution of tilted, incomplete, nonzero-eccentricity FCDs into

upright, complete, zero-eccentricity TFCDs that need no curvature walls to fill space with smectic layers.

## Discussion

Using water as a substrate for a hybrid-aligned LC film, we have demonstrated two new pathways of defect transformation at the N-SmA transition: a direct transformation of $+1$ converging boojums to TFCDs, and a sequence of morphological changes from HAN to N stripe undulations, then to rows of incomplete elliptic–hyperbolic FCDs and finally to a packing of TFCDs in the rest of the sample. The original boojum network strongly influences the geometry of the subsequent states. The orientations of the FCDs reveal the sense of rotation of the director field along the vertical direction in the N phase—distinguishing, for example, converging and diverging $+1$ boojums—that is otherwise difficult to determine using optical microscopy. Through linear stability analysis and Landau-de Gennes numerical modelling, we have provided an explanation for the N stripe instability as the elastic constant ratio $K_3/K_1$ diverges near $T_{NA}$. Moreover, we have rationalized the transition from N stripes to smectic FCD rows by illustrating the similarity in their director fields. These results reveal new connections between the defects of the N and SmA phases, and demonstrate a strong history dependence in the geometries of the defect configurations. Such temperature-controlled transformations and orientational memory provide avenues for switchable behaviour in applications that exploit the interactions of light, colloidal particles or nanoparticles with liquid crystalline materials.

## Methods

**Sample preparation.** As a reservoir, we used a silicon wafer that has a circular hole with a 50 μm depth and 4 mm diameter. To reduce the meniscus of the water, we treated the $O_2$ plasma to minimize the meniscus before we dropped the water into the reservoir, and we casted 8CB (Sigma-Aldrich) on the water filled in the reservoir. The LC film thickness is $\sim 1.5$ μm that varies with an amount of LC materials. The temperature control to induce the phase transition of LCs was performed on a heating stage (LINKAM LTS420) using a temperature controller (LINKAM TMS94).

**Optical characterization.** All the experimental results are directly shown by POM with and without a retardation plate.

**Linear stability analysis of nematic stripe distortions.** Here we calculate the director field of the uniformly distorted HAN state, and then perform linear stability analysis to predict the appearance of stripe deformations. We first seek the reference nematic (N) configuration before the onset of stripe undulations. The Frank elastic energy is

$$F_{elastic} = \frac{1}{2}\int dV \{K_1(\nabla\cdot\mathbf{n})^2 + K_2(\mathbf{n}\cdot(\nabla\times\mathbf{n}))^2 + K_3((\mathbf{n}\cdot\nabla)\mathbf{n})^2 \\ - 2K_{24}\nabla\cdot[\mathbf{n}(\nabla\cdot\mathbf{n}) - (\mathbf{n}\cdot\nabla)\mathbf{n}]\}. \quad (2)$$

The saddle-splay divergence term can be transformed into a surface integral that in our thin cell of (constant) thickness $H$ becomes

$$-K_{24}\{\hat{\mathbf{z}}\cdot[\mathbf{n}(\nabla\cdot\mathbf{n}) - (\mathbf{n}\cdot\nabla)\mathbf{n}]_{z=H} - \hat{\mathbf{z}}\cdot[\mathbf{n}(\nabla\cdot\mathbf{n}) - (\mathbf{n}\cdot\nabla)\mathbf{n}]_{z=0}\}. \quad (3)$$

We will describe the director field with the usual polar and azimuthal angles $\theta$ and $\phi$, both of which may be functions of spatial coordinates: $\mathbf{n} = (\sin\theta\cos\phi, \sin\theta\sin\phi, \cos\theta)$. In the HAN state, before the onset of stripes, the director has constant azimuthal component, so we can set $\phi = 0$, and $\theta$ depends only on $z$: $\mathbf{n} = (\sin\theta(z), 0, \cos\theta(z))$. This director field has no twist or saddle-splay, and the splay and bend terms in $F$ give a bulk elastic energy density of $\frac{1}{2}K_1\theta'(z)^2[\sin^2\theta + k_3\cos^2\theta]$, where $k_3 \equiv K_3/K_1$. The corresponding Euler–Lagrange equation is $0 = \theta''(z)[1 - \kappa\sin^2\theta] - \theta'(z)^2[\kappa\sin(2\theta)]$ where $\kappa \equiv 1 - k_3^{-1}$. This Euler–Lagrange equation is solved by $\theta(z) = E^{-1}(az + b, \kappa)$, the inverse function of the elliptic integral of the second kind with parameter $\kappa$[47]. With boundary conditions $\theta(0) = \theta_1$, $\theta(H) = \theta_2$, we have

$$E(\theta(z),\kappa) = (z/H)(E(\theta_2,\kappa) - E(\theta_1,\kappa)) + E(\theta_1,\kappa). \quad (4)$$

For $K_3 = K_1$, a simplifying assumption commonly employed deep in the N phase, we have $\kappa = 0$, and since $E(u, 0) = u$, the solution is the well-known linear form $\theta(z) = (z/H)(\theta_2 - \theta_1) + \theta_1$. As we approach the N–SmA transition, $k_3^{-1} = K_1/K_3 \to 0$ so $\kappa \to 1$. Noting that $E(u, 1) = \sin u$, the solution becomes $\theta(z) = \arcsin[(z/H)(\sin\theta_2 - \sin\theta_1) + \sin\theta_1]$.

We now assume infinitely strong anchoring at the homeotropic interface, $\theta(H) = \theta_2 = 0$, which is probably appropriate close to $T_{NA}$ where homeotropic anchoring strengths increase by orders of magnitude[38]. Then in the large $K_3/K_1$ limit,

$$\theta(z) = \arcsin\left[s_1\left(1 - \tfrac{z}{H}\right)\right] \equiv \theta_0(z), \tag{5}$$

where $s_1 \equiv \sin\theta_1$. To keep the calculation tractable, we ignore the corrections to $\theta_0(z)$ in small $K_1/K_3$, the first such term being

$$+ \tfrac{1}{2}\frac{K_1}{K_3}\frac{\left(\left(1 - \tfrac{z}{H}\right)\operatorname{arctanh}[s_1] - \operatorname{arctanh}\left[\left(1 - \tfrac{z}{H}\right)s_1\right]\right)}{\sqrt{1 - \left(s_1(1 - z/H)\right)^2}}. \tag{6}$$

After integrating the elastic energy density of this $\theta_0(z)$ profile along $z$ through the thickness $0 \leq z \leq H$, we obtain a reduced elastic free energy per unit area $\bar{f}_{elastic} = (2H/K_1)f_{elastic} = s_1 \operatorname{arctanh} s_1 + (k_3 - 1)s_1^2$. To this we now add the degenerate planar anchoring energy per unit area from the substrate, using the Rapini–Papoular form, $\bar{f}_{surface} = (2H/K_1)f_{surface} = h\cos^2\theta_1 = h(1 - s_1^2)$ where $h = HW/K_1$ is the ratio of the thickness $H$ to the anchoring extrapolation length $L = K_1/W$. In the undistorted homeotropic state, the only energy per unit area is $\bar{f}_{surface} = h$. Therefore, the total reduced energy per unit area in the homogeneous HAN state, relative to the undistorted homeotropic state, is $\bar{f}_{tot} = \bar{f}_{elastic} + \Delta\bar{f}_{surface} = s_1 \operatorname{arctanh} s_1 + (k_3 - h - 1)s_1^2$. Minimizing $\bar{f}_{tot}$ with respect to $s_1$, we find $0 = s_1\left(1 - s_1^2\right)^{-1} + \operatorname{arctanh}(s_1) + 2(k_3 - h - 1)s_1$. If $h \leq k_3$, the only solution is $s_1 = 0$, the uniform homeotropic state. Nonzero solutions for $s_1$ exist if and only if $h > k_3$, in which case the HAN state is stable over the undistorted homeotropic state. Because the HAN state does not revert to the homeotropic state in experiment as the temperature is cooled towards $T_{NA}$, it follows that the degenerate planar anchoring strength $W$ increases during cooling at least as fast as $k_3 = K_3/K_1$, so that $h - k_3 = (HW - K_3)/K_1$ remains positive.

To calculate the energy of an infinitesimal stripe perturbation to the reference state following refs 38,39, we assume that the director has polar and azimuthal angles of the form $\theta(y, z) = \theta_0(z) + \vartheta\sin(qy)$, $\phi(y, z) = \varphi\sin(qy + \beta)$, where $\vartheta, \varphi \ll 1$, $q$ is the stripe wavenumber and $\beta$ is an unknown phase shift. The assumption that $\vartheta$ and $\varphi$ are independent of $z$ is not justified a priori, but we do not venture here to solve the coupled differential equations that otherwise arise for $\vartheta(z)$ and $\varphi(z)$. Rather than numerically solving those Euler–Lagrange equations, we compare the analytic results here with the results of the numerically calculated configurations obtained using the Q-tensor approach. A phase shift $\beta = \pi/2$ between the $\theta$ and $\phi$ undulations is found generally to give the most energetically favourable stripes and the broadest $k_3$ interval of stripe instability, and hence we assume $\beta = \pi/2$ hereafter.

The reduced energy per unit area is

$$\bar{f}_{tot} = \frac{2H}{K_1}f_{tot} = \frac{\chi}{2\pi}\int_0^H dz \int_0^{\frac{2\pi}{q}} dy\left[(\nabla\cdot\mathbf{n})^2 + k_2(\mathbf{n}\cdot(\nabla\times\mathbf{n}))^2 + k_3((\mathbf{n}\cdot\nabla)\mathbf{n})^2\right]$$
$$+ \frac{qh}{2\pi}\int_0^{\frac{2\pi}{q}} dy\cos^2[\theta(z=0)] + \frac{2\chi k_{24}}{2\pi}\int_0^{\frac{2\pi}{q}} dy\,\hat{\mathbf{z}}\cdot[\mathbf{n}(\nabla\cdot\mathbf{n}) - (\mathbf{n}\cdot\nabla)\mathbf{n}]_{z=0}, \tag{7}$$

where $\chi \equiv qH$ and $k_i \equiv K_i/K_1$. Expanding to second order in $\vartheta$ and $\varphi$, and performing the integrations, gives an energy difference relative to the reference state ($\vartheta = \varphi = 0$) of $\Delta\bar{f}_{tot} = A\vartheta^2 + B\varphi^2 - 2C\vartheta\varphi$, where $A = -(h/2) - (k_3 - h - 1)s_1^2 + \tfrac{1}{2}k_2\chi^2 + \tfrac{1}{2}(k_3 - 1)s_1 \operatorname{arctanh}[s_1]$, $B = \tfrac{1}{6}s_1^2\chi^2$, $C = \tfrac{1}{4}(k_3 + 1 - 4k_{24})s_1^2\chi$.

The onset of the instability of the homogeneous HAN state to the stripe state is marked by the determinant of the matrix of second derivatives of $\Delta\bar{f}_{tot}$ (with respect to $\vartheta$ and $\varphi$) changing from positive to negative, that is, $0 = AB - C^2$, which gives

$$0 = \frac{1}{12}s_1^2\chi^2\Big[k_2\chi^2 + (k_3 - 1)s_1 \operatorname{arctanh}[s_1] - h$$
$$+ \left(2(h - k_3 + 1) - \frac{3}{4}(k_3 - 4k_{24} + 1)^2\right)s_1^2\Big]. \tag{8}$$

The homogeneous HAN state is unstable when the right-hand side is negative, requiring the factor in square brackets to be negative. Because $\chi$ appears there only in the positive quantity $k_2\chi^2$, the instability exists for $\chi^2$ smaller than the critical value $\chi_\star^2$ that makes the term in square brackets vanish:

$$\chi^2 < \chi_\star^2 = \frac{h - (k_3 - 1)s_1 \operatorname{arctanh}[s_1] + \left(\frac{3}{4}(k_3 - 4k_{24} + 1)^2 - 2(h - k_3 + 1)\right)s_1^2}{k_2}. \tag{9}$$

The numerator must be positive in order for the stripe solution to exist. The vanishing of the numerator implicitly defines the critical value $k_3^\dagger$ of $k_3$, above which stripes appear as the N phase is cooled (Fig. 4b). Unfortunately, a closed-form expression for $k_3^\dagger$ cannot be written because $s_1$ depends nontrivially on $k_3$. The critical stripe wavelength corresponding to $\chi_\star$ is $\lambda_\star = 2\pi H/\sqrt{\chi_\star^2}$ (Fig. 4a), and the allowed stripe wavelengths are $\lambda \geq \lambda_\star$. The predicted stripe wavelength diverges as $k_3$ approaches $h$, where, as noted above, the HAN configuration would become unstable to the uniform homeotropic configuration.

Finally, we note that the bend energy-driven stripe instability as demonstrated in Fig. 4c requires $K_3 > 4K_{24}$. If $K_3 < 4K_{24}$, then a different stripe instability occurs, with a cost in bend energy and driven by a decrease in the saddle-splay energy term.

**Numerical modelling.** Numerical modelling of the N stripe instability is conducted using the Landau-de Gennes numerical modelling technique[49]. The N configuration is represented by a traceless, symmetric, rank-3 tensor $Q(\mathbf{r})$ that, in a uniaxial N, is related to the director $\mathbf{n}$ by $Q_{ij} = \tfrac{3}{2}S(n_in_j - \tfrac{1}{3}\delta_{ij})$. Here $S$ is the N degree of order. The following nondimensionalized Landau-de Gennes free energy is minimized over $Q(\mathbf{r})$:

$$F_{LdG} = \int d^3r\,(f_{phase} + \tilde{L}f_{elastic}) + \int dS f_{anch}, \tag{10}$$

$$f_{phase} = -\frac{1}{2}\operatorname{tr}(Q^2) + \frac{1}{3}\tilde{B}\operatorname{tr}(Q^3) + \frac{1}{4}\tilde{C}(\operatorname{tr}(Q^2))^2, \tag{11}$$

$$f_{elastic} = \frac{1}{2}\frac{\partial Q_{ij}}{\partial x_k}\frac{\partial Q_{ij}}{\partial x_k} + \frac{1}{2}\ell_2\frac{\partial Q_{ij}}{\partial x_j}\frac{\partial Q_{ik}}{\partial x_k} + \frac{1}{2}\ell_3 Q_{ij}\frac{\partial Q_{kl}}{\partial x_i}\frac{\partial Q_{kl}}{\partial x_j}$$
$$- \ell_{24}\left[\partial_iQ_{ij}\partial_kQ_{jk} - \partial_iQ_{jk}\partial_kQ_{ij}\right]. \tag{12}$$

Here, the first integral in $F_{LdG}$ integrates the phase and elastic free energy densities over the bulk, while the second integral contains the surface anchoring contribution from the bottom (water) substrate. The top (air) interface is assumed to impose infinitely strong homeotropic anchoring, fixing $Q(\mathbf{r})$ there in a uniaxial configuration with $\mathbf{n}\|\hat{\mathbf{z}}$. The 'phase' free energy parameters $\tilde{B}$ and $\tilde{C}$ are respectively taken to be $-12.3$ and $10.1$, values commonly assumed in modelling 5CB[49], giving preferred bulk degree of order $S = S_0 = 0.533$. $\tilde{L}$ is a dimensionless parameter controlling the importance of the elastic free energy compared with the phase free energy; we set this to a small value $\tilde{L} = 0.1$ so that the N is approximately uniaxial everywhere, enabling comparison to the predictions of the Frank free energy. Because of the strong uniaxiality and the absence of defects, the exact values of $\tilde{B}$ and $\tilde{C}$ are not important. All lengths are implicitly in units of the simulation mesh spacing, such that $\tilde{L}$ is proportional to the ratio of the mesh spacing to the N correlation length.

The Frank elastic constant ratios $k_i = K_i/K_1$ are related to the Q-tensor elastic constant ratios $\ell_i$ by $\ell_2 = -6(k_2 - 1)/(k_3 + 3k_2 - 1)$, $\ell_3 = (2/S_0)(k_3 - 1)/(k_3 + 3k_2 - 1)$, $\ell_{24} = 6(k_{24} - \tfrac{1}{2}k_2)/(k_3 + 3k_2 - 1)$. In the anchoring energy density $f_{anch}$, degenerate planar anchoring at the water interface is modelled using the orientational anchoring component of Fournier and Galatola's anchoring potential[54], $f_{anch} = W_1\left(\tilde{Q}_{\alpha\beta} - \tilde{Q}_{\alpha\beta}^\perp\right)\left(\tilde{Q}_{\alpha\beta} - \tilde{Q}_{\alpha\beta}^\perp\right)$, where $\tilde{Q}_{\alpha\beta} = Q_{\alpha\beta} + \tfrac{1}{2}S_0\delta_{\alpha\beta}$, $\tilde{Q}_{\alpha\beta}^\perp = P_{\alpha\gamma}\tilde{Q}_{\gamma\delta}P_{\delta\beta}$, using the projection operator $P_{\alpha\beta} = \delta_{\alpha\beta} - v_\alpha v_\beta$ with substrate normal $\hat{\mathbf{v}} = \hat{\mathbf{z}}$.

In the uniaxial limit, if the director makes an angle $\theta$ with the surface normal, $f_{anch}$ reduces to $f_{anch} = \tfrac{9}{8}S_0^2W_1\cos^2\theta(3 - \cos 2\theta)$. The dependence on $\theta$ is different from that of the Rapini–Papoular form $f_{anch} = \tfrac{1}{2}W\cos^2\theta$, but the two forms agree near $\theta = \pi/2$ if we make the correspondence $W = 9S_0^2W_1$. In terms of the reduced cell thickness $h = HW/K_1$, $W_1 = hK_1/(9S_0^2H) = (h\tilde{L})/(4H)(2 + \ell_2 - S_0\ell_3)$.

The total free energy $F_{LdG}$ is then minimized using a finite difference scheme on a regular square mesh, using a conjugate gradient algorithm from the ALGLIB package (http://www.alglib.net).

As predicted by the Frank elasticity theory approach, the nematic is uniformly homeotropic for $h \leq k_3$, and adopts a homogenous HAN deformation if $k_3$ is smaller than both $h$ and $k_3^\dagger$, the critical elastic constant ratio for the onset of stripes. When $k_3^\dagger < k_3 < h$, stripe undulations appear in the energy-minimizing director field (Fig. 5c,d,f,g). The stripe wavelength generally decreases with increasing $k_3$ and is close to $\lambda_\star$ predicted from the linear stability analysis (Fig. 4a). The critical ratio $k_3^\dagger$ and the stripe wavelength $\lambda$ at a given $k_3$ depend on $k_2$, $k_{24}$ and $h$.

**Data availability.** The data that support the findings of this study are available from the corresponding author on request.

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

## Acknowledgements

This work was supported by a grant from the National Research Foundation (NRF) and funded by the Korean Government (MSIP) (2014M3C1A3052567 and 2015R1A1A1A05000986). D.A.B. was supported by Harvard University through a George F. Carrier Fellowship.

## Author contributions

M.-J.G. and D.A.B. contributed equally to this work. M.-J.G. and D.K.Y. designed the research; M.-J.G. performed experimental work; D.A.B. carried out theoretical and numerical work; M.-J.G., D.A.B. and D.K.Y. analyzed results and wrote the manuscript.

## Additional information

**Competing interests:** The authors declare no competing financial interests.

**Publisher's note**: 

