## [Peer Review File · Nature Communications]

Reviewers' comments:

Reviewer #1 (Remarks to the Author):

This manuscript examines in detail the structural changes that occur in topological defects as a liquid crystal undergoes a phase transition from a nematic to a smectic state. This paper is important in several respects. First, defects in liquid crystals play a central role in their structure, thermodynamic, and dynamic properties. In fact, one could even say that defects "define" liquid crystals. And yet, very little is known about the details of defects. From a theoretical point of view, models simply assume that defects are singularities where the order changes abruptly and the free energy creates mathematical problems. From an experimental point of view, defects are simply viewed as interesting objects that distort light in particular ways. Few researchers (if any), have in fact characterized defects with molecular level of detail, either experimentally or computationally, and few studies have followed the structure and dynamics of defects as a material (and the underlying defects) undergoes a phase transition. I therefore would encourage publication of this article, but only after the authors address the following questions and concerns:

1) A first concern pertains to the theoretical treatment of the transition. The authors adopt a model for nematic materials to describe a transition to a smectic phase. They do so by manipulating the relative magnitude of the elastic constants. There are, however, better models with which to describe smectic materials. The authors should consider using such models, or at least explain and justify their choice in detail.

2) The authors' discussion of the literature is wanting. An account of recent studies of defect structure evolution during phase transitions is seriously incomplete and must be improved. Important recent works by Zummer and co-workers, Cates and co-workers, or de Pablo and co-workers, for example, should be cited. Those works include both experiments and numerical studies. The results of the authors should be framed in the context of those reported in those recent reports.

3) The discussion of the N stripe instability reported by the authors, which is based on their theoretical calculations, is highly phenomenological and speculative. Few molecular or thermodynamic arguments are offered. It does not have to be that way. A more detailed discussion of how several terms in the free energy contribute to the observed changes, why and how they arise for 8CB, is completely accessible to the authors and would strengthen the manuscript considerably.

To be sure, this is really a fine piece of work, but it can and it should be improved considerably. I look forward to the authors' responses to my concerns.

Reviewer #2 (Remarks to the Author):

Prof. Yoon's team is describing transformation of defects upon nematic-smectic transition. Twist and bend of director are incompatible with smectic ordering. As a result, the textures change considerably with up reaching the transition temperature from nematic to smectic phase. The substantial change of these constants while still in a nematic phase alters the textures of the director field even before the transition and then further when in the smectic phase. The kinetics is such that the smectic textures are partly defined by the ones in the nematic phase. The paper could be of a broad interest and could

make a strong impact. I see no obvious deficiencies in this work in terms of interpretation of the findings. Authors did a good job writing this paper clearly. In terms of overview, to set a broader platform, perhaps things like Kosterlitz-Tholues transitions mediated by defects could be mentioned. Also, authors mention Prof. Dhara's work in references but not an earlier paper PRE 72, 031704 (2005) with particles in a smectic system. Besides these minor things, the paper can be published.

Reviewer #3 (Remarks to the Author):

The authors address the formation of focal conic domains in hybrid confinement of a liquid crystal. The carefully performed experiments with a theoretical analysis of pre-transitional stripe patterns and of the evolution of focal conics add some novel information on these transitional phenomena. Its presentation is not oriented toward a broad audience. So it will probably attract only some specialist in the field. Therefore, the paper does not fit in the Nature Communications. With changes mentioned below it can be published in the PRE or Liquid Crystals or with even more adaptations in the Scientific Reports.

Here is the list of my particular remarks:

Title

The title is too general!

Line 44

Ref2 does not seem to be the best choice for that spot.

Lines 56-59

Reorganization of the ordering field at the N-SmA transition is probably accompanied by backflow. How relevant it can be?

Line 63

A comment that surface tension is high enough to prevent nematic surface defects to perturb flatness of the interface would not hurt.

Lines 83-89

What is the thickness of the drop and how it is controlled? What is the diameter of the drop? What are anchoring conditions on the lateral boundary?

Line 136 -140

The boojum sign selection and their naming differ in the literature. See recent papers by Jiang et al. LC 2016 and Kos et al. Soft Matter 2016! I believe that it is not the best for this paper to go against the well spread use of signs of hedgehogs where a radial 3D director field corresponds to +1 and a hyperbolic to -1. The terms used for the director field characterization "converging" and "diverging" are OK. One should have in mind that most of the readers are not specialist in the topology.

Lines 141-161

The discussion of FCDs and TFCDs without an introduction to focal conics comes too early it should be after the paragraph where nematic stripes are explained.

Lines 163-207

The stability analysis first needs a qualitative explanation and illustration what director does to reduce high cost of the bent and to yield two distinct patterns of stripes for two types of boojums. It would

useful to as well explain how relevant is a preexisting director field distortion for the appearance of the spontaneous periodic deformation. The rest of the stability analysis has too many formulas written in the text what makes it less comprehensible.

Lines 209 – 255

The paragraph “Geometrical Discussion of the Stripe-FCD Transition” is rather incompressible for readers that are not experts in the field. A kind of introduction to FCDs and TFCDs is needed for the start than text shifted from above should be incorporated. Further the discussion of the transition in textures occurring by entering SmA phase and by further cooling should be illustrated by additional graphics.

Lines 290-400

In the methods the segment on the stability analysis is far too extensive. Mostly it should be incorporated in Supplementary materials.

Figs 1-3

Some parts repeat but some illustrations are missing as mentioned in the remarks above.

Fig4c

The curve in the figure shows increase of the distortion wave length for very large K_3 . Does this mean that linear analysis breaks?

Supplementary materials:

Fig S2 should appear in the main text!

REVIEWERS' COMMENTS:

Reviewer #1 (Remarks to the Author):

The authors have addressed all of my comments and concerns. The paper is now suitable for publication in Nature Communications.

Reviewer #3 (Remarks to the Author):

The authors have mostly followed remarks of referees and have substantially improved the manuscript. Now it mostly conforms to Nature Communications. I also like the new title! Nevertheless, there are still few minor points that need attention before publication.

- My comment in the previous report "A comment that surface tension is high enough to prevent nematic surface defects to perturb flatness of the interface would not hurt." was probably misunderstood. For general audience it is good to mention that anchoring energies are weak compared to the surface tension.

- The justification of the use "converging" and "diverging" in place of "radial" and "hyperbolic" for +1 boojums is convincing. Nevertheless further interesting questions appear that could help to further improve the paper. Is there another method to distinguish between "converging" and "diverging" +1 boojums besides observing a direct formation of TFCD? Are also -1 boojums of two kinds depending on their twisted structure? It would be good to schematically show their structure and stress why they cannot directly lead to a TFCD. It would be good to stress how the number of +1 & -1 boojums is restricted by topology! How is the relative number of "converging" and "diverging" +1 boojums restricted?

- The theoretical analysis in the revised main text is much easier to follow and comprehend. Now the theoretical part of Methods is large and my personal believe is that most of the details should go to the Supplement. Nevertheless, I leave this point to the editor.

- The references 10 and 29 are the same! From their position in text, I would expect that 29 should cover the knotted colloid work by the same group.

Reviewers' comments:

Reviewer #1 (Remarks to the Author):

This manuscript examines in detail the structural changes that occur in topological defects as a liquid crystal undergoes a phase transition from a nematic to a smectic state. This paper is important in several respects. First, defects in liquid crystals play a central role in their structure, thermodynamic, and dynamic properties. In fact, one could even say that defects "define" liquid crystals. And yet, very little is known about the details of defects. From a theoretical point of view, models simply assume that defects are singularities where the order changes abruptly and the free energy creates mathematical problems. From an experimental point of view, defects are simply viewed as interesting objects that distort light in particular ways. Few researchers (if any), have in fact characterized defects with molecular level of detail, either experimentally or computationally, and few studies have followed the structure and dynamics of defects as a material (and the underlying defects) undergoes a phase transition. I therefore would encourage publication of this article, but only after the authors address the following questions and concerns:

A0. Thank you for the consideration of our work, we do our best to revise the manuscript as recommended.

Q1. A first concern pertains to the theoretical treatment of the transition. The authors adopt a model for nematic materials to describe a transition to a smectic phase. They do so by manipulating the relative magnitude of the elastic constants. There are, however, better models with which to describe smectic materials. The authors should consider using such models, or at least explain and justify their choice in detail.

A1. We agree with the reviewer that nematic free energy expressions (Frank-Oseen, Landau-de Gennes) do not, on their own, adequately model the smectic phase. However, we use these models only to calculate a stripe instability that, we believe, occurs within the nematic phase, with temperature slightly above T_{NA} . This regime is characterized by diverging ratios of nematic elastic constants before the appearance of smectic density modulations. Under this assumption, we believe the nematic free

energy expressions are appropriate to describe the stripe instability. Precedent in the literature for use of nematic free energy formulations to study effects just above the nematic-smectic transition include Refs. 12, 20, and 21. We do not attempt to use these models to calculate the energetics of the FCD state, as certainly smectic order has appeared by then.

The commonly accepted theoretical approach to the nematic-smectic A transition is to add to the Frank-Oseen free energy a Ginzburg-Landau expansion in a smectic phase field, leading to the celebrated analogy with superconductors. However, we are not familiar with any previous work that has successfully applied this type of model to configurations as complicated as the incomplete, tilted, elliptic-hyperbolic focal conic domain rows that appear in our system upon cooling below the transition temperature. Analytical approaches seem intractable, and even computational approaches employing this model in 3D face notorious convergence challenges due to the long-ranged effects of perturbations in the smectic density waves. So, while this is undoubtedly the most complete approach, we do not believe it to be practical for modeling our system.

An alternative theoretical approach often used in the study of FCDs is to assume that smectic order is perfectly obeyed except on low-dimensional defects. Then, the free energy is a sum of defect core energies and layer bending energies (similar to the Helfrich model of membranes). In this framework, the energy of a complete FCD was calculated by Kleman and Lavrentovich, *Phys. Rev. E*, 61:1574, 2000. In principle, similar but less elegant calculations could be used to compute the energies of the incomplete FCDs in our system, but many assumptions would have to be made about the structure of the defects between FCDs and those defects' energetic costs. We think such calculations could well be worthwhile in future work. However, for the present work, we fear that such calculations would add significantly to the paper's length, technicality, and amount of speculation. Also, the available experimental data would not be able to distinguish between different defect models. We therefore choose in the present work to give a geometrical, qualitative explanation of the connection between stripes and FCDs, leaving the door open for more quantitative studies in future work.

Q2. The authors' discussion of the literature is wanting. An account of recent studies of defect structure evolution during phase transitions is seriously incomplete and must be improved. Important recent works by Zummer and co-workers, Cates and co-workers, or de Pablo and co-workers, for example, should be cited. Those works include both experiments and numerical studies. The results of the authors should be framed in the context of those reported in those recent reports.

A2. As recommended by the reviewer, we have added citations to several works on defects at the nematic-smectic phase transition, many of them recent, in the Introduction section. We have also added text in the Introduction to clarify the context of our results considering the recent findings about defects at the N-SmA transition in other systems.

Q3. The discussion of the N stripe instability reported by the authors, which is based on their theoretical calculations, is highly phenomenological and speculative. Few molecular or thermodynamic arguments are offered. It does not have to be that way. A more detailed discussion of how several terms in the free energy contribute to the observed changes, why and how they arise for 8CB, is completely accessible to the authors and would strengthen the manuscript considerably.

A3. We have added, in Fig. 4c and the associated discussion in the text, an examination of how the various components of the free energy change with stripe amplitude. It is revealed that the stripes increase all energy components except the bend term, which decreases. Thus, as bend becomes increasingly costly upon approaching the phase transition temperature (a property common to the N-SmA transitions in any material), the stripes eventually become energetically favorable. Furthermore, in the discussion of the newly added Figure 5, we explore how the numerically calculated stripe director field structure expands regions of splay distortion and causes regions of bend distortion to shrink, in the plane of the substrate (Fig. 5d,g).

We also note, in the Methods section, that when the saddle-splay elastic constant exceeds one-quarter of the bend elastic constant, there is a stripe instability driven by the saddle-splay energy term rather than the bend energy term, although we do not

believe this to apply to the experiment or numerics.

Reviewer #2 (Remarks to the Author):

Prof. Yoon's team is describing transformation of defects upon nematic-smectic transition. Twist and bend of director are incompatible with smectic ordering. As a result, the textures change considerably with up reaching the transition temperature from nematic to smectic phase. The substantial change of these constants while still in a nematic phase alters the textures of the director field even before the transition and then further when in the smectic phase. The kinetics is such that the smectic textures are partly defined by the ones in the nematic phase. The paper could be of a broad interest and could make a strong impact. I see no obvious deficiencies in this work in terms of interpretation of the findings. Authors did a good job writing this paper clearly.

A0. Thank you so much for the kind consideration of our work.

Q1. In terms of overview, to set a broader platform, perhaps things like Kosterlitz-Thouless transitions mediated by defects could be mentioned. Also, authors mention Prof. Dhara's work in references but not an earlier paper PRE 72, 031704 (2005) with particles in a smectic system. Besides these minor things, the paper can be published.

A1. To emphasize the relevance of our work to a broad readership, we describe the connection between our area of research and the Kosterlitz-Thouless transition in the Introduction section. We thank the reviewer for this suggestion. Regarding the earlier reference PRE 72, 031704 (2005) mentioned by the reviewer: We thank the reviewer for bringing this work to our attention. However, we focus on our Introduction specifically on works describing defects at the nematic-smectic transition, whereas this earlier work describes defects only within the smectic phase. Defects around colloids in the SmA phase have been explored by others as well, including

prominently Blanc and Kleman, *Eur. Phys. J. E*, 4 241–251 (2001). We choose not to cite such works, not out of lack of appreciation for their importance, but simply to keep our manuscript focused on the phase transition.

Reviewer #3 (Remarks to the Author):

The authors address the formation of focal conic domains in hybrid confinement of a liquid crystal. The carefully performed experiments with a theoretical analysis of pre-transitional stripe patterns and of the evolution of focal conics add some novel information on these transitional phenomena. Its presentation is not oriented toward a broad audience. So it will probably attract only some specialist in the field. Therefore, the paper does not fit in the Nature Communications. With changes mentioned below it can be published in the PRE or Liquid Crystals or with even more adaptations in the Scientific Reports.

A0. We sincerely appreciate your deep comments, which have helped us to improve our manuscript in revision. Our replies to each comment are marked in red. Furthermore, to improve accessibility to a broad readership as recommended by all reviewers, we have substantially revised the Introduction (with added discussion and more literature citations) and the theoretical sections (with fewer equations and more illustrations) of the main text.

Here is the list of my particular remarks:

Q1. Title

The title is too general!

A1. We revised the title to ‘Morphogenesis of liquid crystal topological defects of during the nematic-smectic A phase transition’.

Q2. Line 44

Ref2 does not seem to be the best choice for that spot.

A2. We changed the ref. 2. to ‘M. V. Kurik and O. D. Lavrentovich, Defects in liquid

crystals: homotopy theory and experimental studies. Sov. Phys. Usp. 31, 196–224 (1988)'.

Q3. Lines 56-59

Reorganization of the ordering field at the N-SmA transition is probably accompanied by backflow. How relevant it can be?

A3. During the phase transition, LC molecules reorganize to form some structures by considering surface anchoring and elastic constants, not back flow. The sentence in lines 56-59 means that strong anchoring at the boundaries can restrict and disturb the gradual molecular reorientation during the phase transition.

Q4. Line 63

A comment that surface tension is high enough to prevent nematic surface defects to perturb flatness of the interface would not hurt.

A4. The sentence does not mean that the strong homeotropic anchoring prevents the surface defects. It means that the antagonistic boundary conditions of air/LC and LC/water make the LC molecules form hybrid alignment.

Q5. Lines 83-89

What is the thickness of the drop and how it is controlled? What is the diameter of the drop? What are anchoring conditions on the lateral boundary?

A5. We estimate the LC thickness is $\sim 1.5 \mu\text{m}$ based on the size of TFCDs and birefringent color of the film. And the diameter of the LC film is approximately equal to the diameter of the reservoir (4 mm), which means we can ignore the lateral boundary condition because all the experiments were performed in the center of the LC droplet. Simply, the dimension of the LC film can be controlled by the amount of the LC material. This information is also mentioned at Methods section. The LC molecules feel homeotropic anchoring from the air interface and planar anchoring from the water interface.

Q6. Line 136 -140

The boojum sign selection and their naming differ in the literature. See recent papers by Jiang et al. LC 2016 and Kos et al. Soft Matter 2016! I believe that it is not the best for this paper to go against the well spread use of signs of hedgehogs where a radial 3D director field corresponds to +1 and a hyperbolic to -1. The terms used for the director field characterization “converging” and “diverging” are OK. One should have in mind that most of the readers are not specialist in the topology.

A6. We believe our labeling of boojum sign is consistent with the literature, including the two references mentioned by the reviewer: The boojum sign is the winding number of the director field in the surface of the boundary, rather than the “hedgehog charge” of the analogous defect in bulk (see for example Fig. 2 in Kos et al, Soft Matter, 2016, 12, 1313). Our nomenclature does differ from convention in our use of the terms “converging” and “diverging” in place of “radial” and “hyperbolic”. Our choice of terminology serves to connect the two +1 boojum structures to the resulting orientation of the hyperbola defects in the smectic focal conic domains. Furthermore, we consider it prudent to avoid using the word “hyperbolic” for boojums since the focal conic domains have hyperbolic defect curves, and we wish to avoid confusing the reader on this point. In deference to the literature convention, however, we have added the following sentence on page 8: “The +1 converging boojum is frequently referred to in the literature as “hyperbolic”, and the +1 diverging boojum as “radial”.”

Q7. Lines 141-161

The discussion of FCDs and TFCDs without an introduction to focal conics comes too early it should be after the paragraph where nematic stripes are explained.

A7. In the second paragraph of the section “Morphological Changes Observed Experimentally at the Phase Transition”, we have added an explanation of the geometry of TFCDs and FCDs. We have also added Fig. 1c, an illustration of the elliptic-hyperbolic FCD, in comparison with the TFCD in Fig. 1b.

Q8. Lines 163-207

The stability analysis first needs a qualitative explanation and illustration what director does to reduce high cost of the bent and to yield two distinct patterns of stripes for two types of boojums. It would useful to as well explain how relevant is a

preexisting director field distortion for the appearance of the spontaneous periodic deformation. The rest of the stability analysis has too many formulas written in the text what makes it less comprehensible.

A8. To illustrate the director field of the stripe configuration in both theory and numerics, we have added Fig. 5. To better explain the energetic cause of the stripes, we have added, in Fig. 4c and the associated discussion in the text, an examination of how the various components of the free energy change with stripe amplitude. It is revealed that the stripes increase all energy components except the bend term, which decreases. Thus, as bend becomes increasingly costly upon approaching the phase transition temperature, the stripes eventually become energetically favorable. Furthermore, in the discussion of the newly added Figure 5, we explore how the numerically calculated stripe director field structure expands regions of splay distortion and causes regions of bend distortion to shrink, in the plane of the substrate (Fig. 5d,g).

We have substantially rewritten the section “Theory of the Nematic Stripe Instability.” Most equations have been moved to the Methods section. Instead, in the main text we provide a qualitative overview of the theoretical calculation and its results, and in the newly added Fig. 5 and Fig. 4c,d we provide visualizations of the stripe state’s director field and calculated energy. We hope that these changes improve ease of reading and understanding while offering sufficient details in the Methods section to fully explain our work.

Regarding the two distinct patterns of stripes for converging and diverging boojums, we have added the following sentence in the discussion of Fig. 2: “Because the stripes form parallel to the N phase’s horizontal \mathbf{n} component, the radial or diamond arrangement of stripes inherits and reveals the +1 (radial) or -1 (hyperbolic) windings of the original boojum configuration.”

Regarding the relevance of a preexisting director field distortion for the appearance of the spontaneous periodic deformation: We cite previous work (Refs. 20, 21) on the appearance of stripe distortions at the N-SmA transition when the N configuration is undistorted. Thus it is not required to have a pre-existing distortion to observe stripes

at the transition. However, importantly, the stripes in our distorted system emerge for energetic reasons quite distinct from those in Refs. 20, 21 because the pre-existing bend in our system becomes increasingly costly as the phase transition is approached. Fig. 4c and associated discussion in the text help to clarify this point.

Q9. Lines 209 – 255

The paragraph “Geometrical Discussion of the Stripe-FCD Transition” is rather incompressible for readers that are not experts in the field. A kind of introduction to FCDs and TFCDs is needed for the start than text shifted from above should be incorporated. Further the discussion of the transition in textures occurring by entering SmA phase and by further cooling should be illustrated by additional graphics.

A9. Please see our response to Question 7. The illustration of a complete FCD in Fig. 1c will facilitate understanding of incomplete FCDs in Fig. 6b,d. In Fig. 6c, we have replaced a theoretical sinusoidal director field with the numerically calculated director field, which more closely matches the FCD structure. We have amended the discussion in the section “Geometrical Discussion of the Stripe-FCD Transition” to better discuss Fig. 6. We also add some discussion of the defect walls between FCDs and between adjacent rows of FCDs. As we write in the revised text, the structure of these defect walls is beyond the scope of the present work and is probably very complicated. However, the important point is that these defect walls become increasingly costly as the smectic order saturates during cooling, driving the continuous transition from FCD rows to a TFCD array (which does not require defect walls).

Q10. Lines 290-400

In the methods the segment on the stability analysis is far too extensive. Mostly it should be incorporated in Supplementary materials.

A10. We thank the reviewer for this recommendation. In this revision, we have focused on moving most equations out of the main text and *into* the Methods section, so that the

explanation of the calculation is self-contained but unobtrusive.

Q11. Figs 1-3

Some parts repeat but some illustrations are missing as mentioned in the remarks above.

A11. We hope that our responses above have answered the reviewer's concerns about missing illustrations. Regarding repetition: Given the complexity of the liquid crystalline structures and the broad readership, we find it useful to repeat certain points. Despite some overlap, Figs. 1-3 each serve a distinct purpose. Fig. 1 provides a general overview of the various morphologies encountered in our manuscript. Fig. 2 reveals new insights from adding the quarter-wave plate, and (in this revision) compares the +1 and -1 boojum configurations. Fig. 3 illustrates the differences between converging and diverging boojums.

Q12. Fig4c

The curve in the figure shows increase of the distortion wave length for very large K_3 . Does this mean that linear analysis breaks?

A12. Yes, in fact the underlying assumptions of the whole linear stability analysis are broken where the stripe wavelength diverges on the right side of the plot in Fig 4a.

This occurs as $K_3/K_1=40=h$ (where $h=HW/K_1$). As noted in the main text's discussion about the linear stability calculation, the homogeneously distorted HAN state, which we take as a reference state before adding stripe perturbations, is itself unstable to the uniform homeotropic state when $h < K_3/K_1$. So, linear stability predicts that the maximum K_3/K_1 at which stripes are allowed coincides with the maximum K_3/K_1 at which the HAN state can appear at all, with or without stripes. We infer that, in experiment, "the degenerate planar anchoring strength W increases during cooling at least as fast as $k_3 = K_3/K_1$, so that $h - k_3 = (HW - K_3)/K_1$ remains positive." We

have added a sentence at the end of Methods section (c) reading, “The predicted stripe wavelength diverges as k_3 approaches h , where, as noted above, the HAN configuration would become unstable to the uniform homeotropic configuration.”

Q13. Supplementary materials:

Fig S2 should appear in the main text!

A13. We moved the Fig. S2 to Fig. 2 e-h as recommended.

Reviewer #3 (Remarks to the Author):

The authors have mostly followed remarks of referees and have substantially improved the manuscript. Now it mostly conforms to Nature Communications. I also like the new title! Nevertheless, there are still few minor points that need attention before publication.

Q1. My comment in the previous report “A comment that surface tension is high enough to prevent nematic surface defects to perturb flatness of the interface would not hurt.” was probably misunderstood. For general audience it is good to mention that anchoring energies are weak compared to the surface tension.

A1. Thank you for this clarification. We have added the following sentence to the introduction: “Because the surface tension dominates over liquid crystalline anchoring energies, the LC-water interface is not significantly perturbed from flatness by director distortions or defects.”

Q2. The justification of the use “converging” and “diverging” in place of “radial” and “hyperbolic” for +1 boojums is convincing. Nevertheless further interesting questions appear that could help to further improve the paper. Is there another method to distinguish between “converging” and “diverging” +1 boojums besides observing a direct formation of TFCD? Are also -1 boojums of two kinds depending on their twisted structure? It would be good to schematically show their structure and stress why they cannot directly lead to a TFCD. It would be good to stress how the number of +1 & -1 boojums is restricted by topology! How is the relative number of “converging” and “diverging” +1 boojums restricted?

A2.

Is there another method to distinguish between “converging” and “diverging” +1 boojums besides observing a direct formation of TFCD?

The orientations of the surrounding elliptic-hyperbolic FCDs reveal whether the +1 boojums are converging or diverging: The FCD hyperbolas are oriented toward the converging +1 boojums, and away from the diverging +1 boojums. This is described in the manuscript. Another distinction, which we will explore in future work, is that at the isotropic-nematic transition, the +1 converging boojum is located at the center of a large NLC domain, whereas the +1 diverging boojum is at a boundary among large NLC domains.

Are also -1 boojums of two kinds depending on their twisted structure?

We observe only one type of -1 boojum. No twisted structure is observed. The converging/diverging distinction for +1 boojums is not relevant for -1 boojums, which are always “converging” along one axis and “diverging” along the perpendicular direction because of the hyperbolic profile in the horizontal plane. (See, e.g., Kos and Ravnik, *Soft Matter*, 2016, Fig 2.)

It would be good to schematically show their structure and stress why they cannot directly lead to a TFCD.

The lack of azimuthal symmetry is the reason why -1 boojums do not directly evolve into TFCDs. Rather than creating a new schematic illustration, we have added a sentence making this point by referring the reader to the experimental image Fig. 2f, which clearly shows through the pattern of stripes that the -1 boojum director field lacks azimuthal symmetry. The added sentence, at the end of the section “Morphological Changes Observed Experimentally at the Phase Transition”, reads: “The -1 boojums also cannot evolve directly into TFCDs because they lack the azimuthal symmetry of TFCDs, as revealed by the diamond-like configurations of the stripes parallel to the original n field (Fig. 2f).”

It would be good to stress how the number of +1 & -1 boojums is restricted by topology! How is the relative number of “converging” and “diverging” +1 boojums restricted?

The topological rules for balancing the number of +1 and -1 boojums is a fascinating topic, and we will explore it in future work on networks of boojums/FCDs at larger scales. However, to make precise statements about topology, one needs to address the boundary conditions; i.e., the conservation of total “boojum charge” requires a measuring loop where there are no defects. In this work, we present a system whose area is large compared with the typical boojum spacing, so that we can study the many defects in the droplet’s interior without worrying about the details of the lateral boundaries. As such, we cannot make any definitive statements about global topological conservation laws.

Energetic considerations would lead one to expect that +1 and -1 boojums are locally balanced in number; our system adds new complexities to that notion, which we are preparing to present in future work. Also, there is a general expectation that the number of converging and diverging +1 boojums is equal, at equilibrium in the nematic phase.

However, in our future work we will show that the +1 boojums are again special at the isotropic-nematic transition. The number of +1 converging boojums initially dominates over the number of other types of boojums. The +1 diverging and -1 boojums originate from collisions between large NLC domains nucleated around converging +1 boojums. Because the counting of defect types is thus a more complex issue than one might at first expect, we hope you will understand our choice to leave these interesting questions to our follow-up work.

Q3. The theoretical analysis in the revised main text is much easier to follow and comprehend. Now the theoretical part of Methods is large and my personal believe is that most of the details should go to the Supplement. Nevertheless, I leave this point to the editor.

A3. We thank you for your suggestions that have helped us to improve the clarity and accessibility of our work. The Editor advises us that the Methods section has no word limit, and requests that we “please incorporate as much of the Supplementary Information into the main paper as possible”. Keeping all the mathematical details in the Methods section of the main text therefore seems to be most consistent with the *Nature Communications* format.

Q4. The references 10 and 29 are the same! From their position in text, I would expect that 29 should cover the knotted colloid work by the same group.

A4. Reference 29 is about the N-SmA transition around colloids. We added it to emphasize the trend of research regarding the N-SmA transition. Reference 10 is a seminal paper but concerns only the nematic phase. We think that the references are proper, even if they are similar to each other.